# METRIC GRAPH KERNELS VIA THE TROPICAL TORELLI MAP

## ABSTRACT

We introduce the first graph kernels for metric graphs via tropical algebraic geometry. In contrast to conventional graph kernels based on graph combinatorics such as nodes, edges, and subgraphs, our metric graph kernels are purely based on the geometry and topology of the underlying metric space. A key characterizing property of our construction is its invariance under edge subdivision, making the kernels intrinsically well-suited for comparing graphs representing different underlying spaces. We develop efficient algorithms to compute our kernels and analyze their complexity, which depends primarily on the genus of the input graphs. Empirically, our kernels outperform existing methods in label-free settings, as demonstrated on both synthetic and real-world benchmark datasets. We further showcase their practical utility with an urban road network classification task.

## 1 INTRODUCTION

Graph kernels are powerful tools for measuring similarities between graphs and networks and have become essential in many machine learning tasks. Traditional graph kernels typically rely on combinatorial aspects of graphs, such as nodes, edges, and subgraphs, with additional information from node and edge labels and attributes (Kriege et al., 2020; Vishwanathan et al., 2010; Borgwardt et al., 2020; Nikolentzos et al., 2021; Neumann et al., 2016). While such kernels have demonstrated success in many domains, they can be limited in scenarios where graphs arise from geometric data, and where labels or attributes are absent or incomparable. Such challenges are intrinsic to *metric graphs* and render existing graph kernels fundamentally inapplicable in this setting. In this paper, we introduce the first kernel framework designed specifically for metric graphs.

Metric graphs are geometric realizations of graphs with a length function on their edges. To develop our metric graph kernels, we leverage the known one-to-one correspondence between metric graphs and *abstract tropical curves* from tropical algebraic geometry (Chan, 2021; Cao & Monod, 2025). This correspondence has been studied in tropical geometry via the *tropical Torelli map* (Chan, 2012; Brannetti et al., 2011), which sends any metric graph to a flat torus. In computational settings and especially machine learning, for a weighted graph with generic length function, we can compute a unique symmetric positive definite (SPD) matrix to represent the flat torus. Using the tropical Torelli map, we can effectively identify weighted graphs with SPD matrices, which contain the intrinsic geometric and topological information on the underlying metric graph.

The space of SPD matrices is central in information geometry and can be endowed with various Riemannian metrics to capture its rich geometric structure (Luo et al., 2021; Zhang et al., 2019; Thanwerdas & Pennec, 2023b). Motivated by insights from both tropical geometry and information geometry, we leverage the *Bures–Wasserstein distance* on the space of postive semi-definite (PSD) matrices and propose the *tropical Torelli–Wasserstein (TTW) kernel*. For simplicity and efficient computation, we also propose the *tropical Torelli–Euclidean (TTE) kernel* based on the Euclidean distance. Our graph kernels are purely based on the geometry and topology of the underlying metric graphs, thus are well-suited for comparing graphs that represent different underlying spaces.

**Contributions.** Our work is the first to propose kernels for metric graphs, specifically, which we achieve by merging tropical geometry and information geometry in the context of machine learning. Our specific contributions are listed as follows:

- We propose new graph kernels for metric graphs, leveraging the tropical Torelli map from tropical geometry. Our graph kernels are invariant under edge subdivisions and naturally extend to well-defined kernels on the underlying space of metric graphs.

- We develop algorithms to compute our metric graph kernels. We show that the computational complexity is dominated by graph *genus*—an algebraic topological measure of graph cycles. We give concrete analyses on different levels of graph sparsity.

- We systematically compare our metric graph kernels with other label-free graph kernels on benchmark datasets. We show that our metric graph kernel outperforms other graph kernels in the absence of node/edge labels.

- We conduct experiments on synthetic and real-world datasets, demonstrating superior performance over existing graph kernels. On synthetic data sets, we test the computational runtime of our metric graph kernels and show consistency with our theoretical analysis. For real-world data, we apply our metric graph kernels to a classification task on local urban road networks.

## 2    FROM GRAPHS TO METRIC GRAPHS

A graph consists of discrete sets of nodes and edges. A *metric graph* is a 1-dimensional metric space that can be realized as the underlying space of a graph. Any graph-based quantity extends well to metric graphs if it is compatible with *edge subdivisions*—a concept we will now introduce mathematically, which will then be motivated and illustrated concretely by road networks.

**Edge Subdivisions.** Let $G = (V, E)$ be a graph and $\ell : E \to \mathbb{R}_+$ be a length function. An *edge subdivision* of $e = [u, v] \in E$ is the following operation on $G$: First, add a new node $w$ to $V$, and then replace $e$ by two edges $e' = [u, w]$ and $e'' = [w, v]$ such that $\ell(e) = \ell(e') + \ell(e'')$. An edge subdivision increases both the number of nodes and number of edges by 1. $G'$ is called a *refinement* of $G$, denoted as $G' \geq G$, if $G'$ can be obtained from $G$ by a sequence of edge subdivisions. By definition, all refinements of $G$ share the same underlying metric graph $|G|$.

Note that refinement introduces a new relation between graphs that differs from the usual notion of subgraph inclusion: If $G'$ is a refinement of $G$, then there exists an injection $V(G) \to V(G')$ and a surjection $E(G') \to E(G)$. However, if $G$ is a subgraph of $G'$, then $V(G) \to V(G')$ and $E(G) \to E(G')$ are both inclusions.

**Definition 1.** *A graph kernel $k$ is called a* metric graph kernel *if $k(G'_1, G'_2) = k(G_1, G_2)$ for any refinements $G'_1 \geq G_1$ and $G'_2 \geq G_2$, that is, $k$ is invariant under graph refinements.*

Existing graph kernels are often based on nodes, edges, and subgraphs, which are not invariant under refinements. Therefore, they fail to be metric graph kernels.

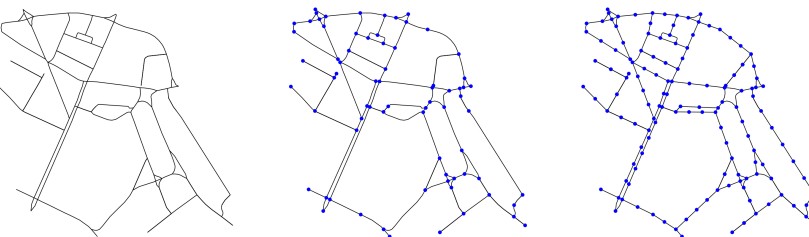

Figure 1: An illustration of a metric graph. The leftmost figure presents a local road network near the Victoria underground (tube) station in London. A graph representing the road network is given in the middle. The rightmost figure presents a refinement by adding artificial landmarks.

**Road Networks.** Typical real-world examples of metric graphs are road networks, which are often modeled using nodes to represent landmarks or intersections and edges to represent roads. However, the essential metric and topological properties of a road network should be independent of its combinatorial representation. For instance, refining a road network by inserting additional landmarks along existing roads should not alter the underlying geometry or structure of the network (Figure 1).

## 3 THE TROPICAL TORELLI MAP FOR WEIGHTED GRAPHS

We first introduce the *tropical Torelli map* for weighted graphs; this is the map we use to send a graph to a matrix. We provide an algorithm for its practical computation, where the uniqueness of the resulting matrix representation is required but challenging to achieve. Under the generic assumption on length functions, our algorithm outputs a unique SPD matrix for any weighted graph. For further details, including related background in tropical geometry and the original definition of the tropical Torelli map on metric graphs, see Appendix A.1.

A concise mathematical formulation of the mapping process for a metric graph to an SPD matrix is as follows: Let $G = (V, E)$ be a graph with a positive weight function $\ell : E \to \mathbb{R}_+$ which here, should be interpreted as the length function on edges. The geometric realization of $G$, denoted as $|G|$, is a 1-dimensional metric space constructed by identifying each edge $e$ as the interval $[0, \ell(e)]$, and gluing all such intervals at endpoints. $|G|$ is also known as a *metric graph*, which is in one-to-one correspondence with abstract tropical curves in tropical geometry (Chan, 2021; Cao & Monod, 2025). The classical tropical Torelli map sends any metric graph to a flat torus or its *tropical Jacobian variety*. For a weighted graph with generic length function, we compute a unique SPD matrix $Q$ to represent the flat torus. Thus, we adapt the tropical Torelli map to weighted graphs by sending any $G$ to the SPD matrix $Q(G)$. We show that $Q(G) = Q(G')$ for any refinement $G'$ of $G$. Hence $Q(G)$ contains the intrinsic geometric and topological information on the underlying metric graph $|G|$.

### 3.1 GRAPH HOMOLOGY

A concept that is central to our construction is the *1-homology group* of a graph. The concept arises from algebraic topology and intuitively corresponds to cycles in a graph. In the following, we give mathematical details on the construction of the 1-homology group of a graph and how we obtain a matrix representation of the graph in terms of 1-cycles via the tropical Torelli map.

We first outline the 1-homology group as follows. Let $G = (V, E)$ be a connected graph with $n$ nodes and $m$ edges. Let $C_0(G; \mathbb{R})$ be the $n$-dimensional vector space spanned by $V$ and $C_1(G; \mathbb{R})$ be the $m$-dimensional vector space spanned by $E$. Fix an arbitrary orientation on $G$. The *boundary map* $\partial : C_1(G; \mathbb{R}) \to C_0(G; \mathbb{R})$ is a linear map given by $\partial([u, v]) = v - u$. The 1-homology group is defined as the kernel of the boundary map $H_1(G; \mathbb{R}) = \ker(\partial)$. Its dimension $g = \dim(H_1(G; \mathbb{R}))$ is called the *genus* of $G$ and satisfies $g = m - n + 1$. Any element $\sigma \in H_1(G; \mathbb{R})$ is called a *1-cycle*. Let $\sigma(e)$ be the coefficient of $e$ in $\sigma$. The *support* of a 1-cycle is the set $\{e \in E : \sigma(e) \neq 0\}$, which forms a cycle in the sense of subgraph.

Next, we describe how to represent the graph as a matrix in terms of 1-cycles. Let $\ell : E \to \mathbb{R}_+$ be a length function. Define an inner product $\mathcal{Q}_G$ on $C_1(G; \mathbb{R})$ by assigning $\mathcal{Q}_G(e_i, e_j) = \delta_{ij}\sqrt{\ell(e_i)\ell(e_j)}$ on the edges and extending bilinearly to the whole space $C_1(G; \mathbb{R})$. Notice that under the inner product $\mathcal{Q}_G$, an element $e_i \in E$ has norm $\sqrt{\ell(e_i)}$ when viewed as a 1-chain in $C_1(G; \mathbb{R})$, in contrast to its length $\ell(e_i)$ when viewed as an edge in $G$ (Ji, 2012). The inner product $\mathcal{Q}_G$ is compatible with edge subdivision: if $e$ is subdivided into $e'$ and $e''$, then

$$\|e' + e''\|_{\mathcal{Q}}^2 = \ell(e') + \ell(e'') = \ell(e) = \|e\|_{\mathcal{Q}}^2.$$

The 1-homology group $H_1(G; \mathbb{R})$ inherits the inner product $\mathcal{Q}_G$ as a closed subspace of $C_1(G; \mathbb{R})$. Fix a 1-cycle basis $\sigma_1, \ldots, \sigma_g$ for $H_1(G; \mathbb{R})$. The inner product $\mathcal{Q}_G$ is represented by a matrix $Q$.

**Definition 2.** *Let $\mathcal{M}_g$ be the space of weighted graphs of genus $g$, and $\mathrm{SPD}(g)$ be the space of $g \times g$ SPD matrices. The* tropical Torelli map *is given by $\mathcal{T} : \mathcal{M}_g \to \mathrm{SPD}(g)$, $G \mapsto Q(G)$.*

### 3.2 MAIN ALGORITHM

We now describe the implementation of the tropical Torelli map to send a graph to its matrix representation as described above. Specifically, we show how to compute a 1-cycle basis for $H_1(G; \mathbb{R})$ and the SPD matrix $Q(G)$.

For a connected graph $G$, we first compute a minimal spanning tree $T$. Fix an arbitrary orientation for $G$. For any edge $e = [u, v] \in G \backslash T$, it generates a 1-cycle $\sigma$ such that $\sigma = e + \sum_{e' \in \gamma} \sigma(e')e'$, where $\gamma$ is the unique path in $T$ connecting $v$ to $u$. Since there are $m - (n - 1) = g$ edges not in

$T$, they generate a 1-cycle basis $\sigma_1, \ldots, \sigma_g$ for $H_1(G; \mathbb{R})$. Construct a $g \times m$ matrix $M$ by setting $M_{ij} = \sigma_i(e_j)$. The matrix $M$ is called the *cycle–edge incidence matrix* (Cao & Monod, 2025).

We reorder the edge set as follows: the first $n-1$ edges $e_1, \ldots, e_{n-1}$ correspond to the minimal spanning tree $T$, while the remaining $g$ edges are sorted in ascending order by length. Then each 1-cycle $\sigma_i$ corresponds to the edge $e_{n-1+i}$. After reordering, the cycle–edge incidence matrix is in the block matrix form $M = [M_T, I_g]$ where $I_g$ is the $g \times g$ identity matrix. Let $L = \mathrm{diag}\{\ell(e_1), \ldots, \ell(e_m)\}$ be the diagonal matrix of edge lengths and $L_T = \mathrm{diag}\{\ell(e_1), \ldots, \ell(e_{n-1})\}$ and $L_g = \{\ell(e_n), \ldots, \ell(e_m)\}$ be its blocks. Then the inner product $\mathcal{Q}_G$ on $H_1(G; \mathbb{R})$ is represented by the matrix

$$Q = MLM^\top = M_T L_T M_T^\top + L_g. \tag{1}$$

For disconnected graphs, we compute $Q$ component-wise and then form a block matrix. We summarize the computation as pseudocode in Algorithm 1.

---

**Algorithm 1:** Computing the tropical Torelli map

---

**Input:** weighted graph $G$
**Output:** matrix $Q$

1  $Q \leftarrow [\,]$
2  **for** *connected component* $G_i = (E_i, V_i)$ **do**
3      compute a minimal spanning tree $T$
4      sort $E_i$ such that $e_1, \ldots, e_{n_i-1}$ are in $T$, and $\ell(e_{n_i}) < \ldots < \ell(e_{m_i})$
5      $M_T \leftarrow \mathrm{zeros}(g_i, n_i - 1)$
6      **for** $j \leftarrow 1$ **to** $g_i$ **do**
7          $[u, v] \leftarrow e_{n_i-1+j}$, find path $\gamma$ from $v$ to $u$ in $T$
8          **for** $k \leftarrow 1$ **to** $(n_i - 1)$ **do**
9              $M_{jk} \leftarrow \gamma(e_k)$
10         **end**
11     **end**
12     $Q \leftarrow \mathrm{diag}\{Q, M_T L_T M_T^\top + L_{g_i}\}$
13 **end**

---

### 3.3 GENERIC LENGTH FUNCTIONS

The output of Algorithm 1, which is our matrix representation of the graph, depends on the choice of minimal spanning trees and ordering of edges, which means that the matrix representation of the graph need not be unique. We now discuss how to achieve the uniqueness of the output. Specifically, we show that the output becomes unique under the genericity assumption of length functions.

**Definition 3.** *Let $G = (V, E)$ be a connected graph. A length function $\ell : E :\to \mathbb{R}_+$ is generic if $G$ has a unique minimal spanning tree $T$ and the lengths of edges are distinct.*

If $G$ is equipped with a generic length function, it admits a canonical orientation. We prove our main theorem on uniqueness and refinement-invariance of the resulting matrix under the tropical Torelli map for weighted graphs in Appendix A.2.

**Theorem 1.** *Let $G = (V, E)$ be a connected graph with a generic length function $\ell$. Under the canonical orientation, the matrix $Q$ computed by Algorithm 1 is unique, and invariant under any refinement of $G$.*

### 3.4 COMPUTATIONAL COMPLEXITY

For a connected graph $G$ with $n$ nodes and $m$ edges, the time complexity to compute the reduced cycle–edge incidence matrix $M_T$ is $O(gn \log n)$. Computing the matrix $Q$ runs in $O(g^2 n)$ time. Therefore, the overall time complexity of Algorithm 1 is $O(gn(g + \log n))$.

The genus $g$ is a fundamental quantity that reflects the topological complexity of a graph and is closely related to its sparsity. We thus propose the following categorizations of graphs into the following *sparsity classes* based on their genus: (i) **Sparse graphs**: $m \asymp n + c$ for some constant $c > 0$; in this case, we have $g = O(1)$, and the total time complexity is $O(n \log n)$. (ii) **Semi-sparse**

**graphs**: $m \asymp cn$ for some constant $c > 0$; in this case, we have $g = O(n)$, and the total time complexity is $O(n^3)$. (iii) **Dense graphs**: $m \asymp n^{1+c}$ for some constant $0 < c \leq 1$; in this case we have $g = O(n^{1+c})$ and the total time complexity is $O(n^{3+2c})$.

Notice that the time complexity ranges from almost linear to quintic, which is in line with the densest scenario for graphs with $O(n^2)$ edges. We remark that the algorithm is particularly fast on sparse graphs, making it well-suited for real-world datasets such as URNs.

## 4 TROPICAL METRIC GRAPH KERNELS

A common approach to constructing a kernel function on a geometric space is to start with a distance function that captures meaningful dissimilarities between elements. Given a distance $d(x, y)$ on the space, a typical strategy is to convert it into a similarity measure using a radial basis function (RBF), $k(x, y) = \exp(-\gamma d^2(x, y))$, where $\gamma > 0$ is a scaling parameter (Que & Belkin, 2016; Scholkopf et al., 1997; Zhu et al., 2020). We now construct our metric graph kernel by first enlarging the embedding space to positive semi-definite (PSD) matrices to enable comparisons between graphs of different genus. Then we construct graph kernels using distances on the space of PSD matrices. We present algorithms to compute these kernels and discuss their computational complexity.

**The Space of Positive Semi-Definite Matrices.**

The tropical Torelli map embeds a graph of genus $g$ as a $g \times g$ symmetric positive definite (SPD) matrix. To compare graphs with different genus, we map the matrices to the same dimension. This is done by modifying the SPD matrices in the following way, which maps from graph space to the space of positive semi-definite (PSD) matrices $\mathrm{PSD}(g_0)$: Given an upper bound $g_0$, we either (i) pad a matrix $Q$ by zero if $\dim(Q) < g_0$, which gives a PSD matrix since $\begin{pmatrix} x^\top & y^\top \end{pmatrix} \begin{pmatrix} Q & 0 \\ 0 & 0 \end{pmatrix} \begin{pmatrix} x \\ y \end{pmatrix} = x^\top Q x \geq 0$, or (ii) extract a submatrix by randomly choosing $g_0$ rows and columns if $\dim(Q) > g_0$, which is stable (see Table 3 in Appendix B.2 for an experimental verification).

Unlike the space of SPD matrices, which forms a smooth manifold that admits various Riemannian metrics, the space of PSD matrices lacks a manifold structure due to the presence of boundary points corresponding to rank-deficient matrices. It is known that $\mathrm{PSD}(g_0)$ is a stratified space which contains $\mathrm{SPD}(g_0)$ as its maximal strata (Thanwerdas & Pennec, 2022; Vandereycken et al., 2013).

### 4.1 TWO TROPICAL KERNELS FOR METRIC GRAPHS

**The Tropical Torelli–Euclidean Kernel.** We begin by introducing a graph kernel based on the Euclidean distance which is a straightforward approach and provides a simple starting point. Let $G_1$ and $G_2$ be two weighted graphs, we define the *tropical Torelli–Euclidean (TTE) kernel* as follows:

$$k_{\mathrm{TTE}}(G_1, G_2) = \exp\big( -\gamma \|Q(G_1) - Q(G_2)\|_F^2 \big),$$

where $\| \cdot \|_F$ is the Frobenius norm of matrices.

Although $k_{\mathrm{TTE}}$ does not take into account the inherent geometry of $\mathrm{PSD}(g_0)$, it has two key advantages: first, the computation is efficient, and second, the kernel induced by the Euclidean distance is known to be positive definite (Haasdonk & Bahlmann, 2004). See Appendix B.1 for a full discussion.

**The Tropical Torelli–Wasserstein Kernel.**

We now turn to proposing a geometric distance on the space of PSD matrices. In the original definition of the tropical Torelli map, a metric graph is sent to a $g$-dimensional flat torus—the quotient space of $\mathbb{R}^g$ by the lattice spanned by the column vectors of $\sqrt{Q}$. Since the flat tori are invariant under orthogonal actions, we consider the following distance on $\mathrm{PSD}(g_0)$:

$$\begin{aligned} W(Q(G_1), Q(G_2)) &= \min_{U_1, U_2 \in O(g_0)} \big\| U_1 \sqrt{Q(G_1)} - U_2 \sqrt{Q(G_2)} \big\|_F \\ &= \min_{U \in O(g_0)} \big\| U \sqrt{Q(G_1)} - \sqrt{Q(G_2)} \big\|_F, \end{aligned} \quad (2)$$

where $O(g_0)$ is the group of orthogonal matrices. It turns out that equation 2 is the *Bures–Wasserstein distance* from information geometry and can be computed from the following closed form (Bhatia

et al., 2019; Thanwerdas & Pennec, 2023a; Altschuler et al., 2021):

$$W^2(Q(G_1), Q(G_2)) = \mathrm{Tr}(Q(G_1)) + \mathrm{Tr}(Q(G_2)) - 2\mathrm{Tr}\left(\sqrt{\sqrt{Q(G_1)}Q(G_2)\sqrt{Q(G_1)}}\right). \quad (3)$$

On $\mathrm{SPD}(g_0)$, the Bures–Wasserstein distance corresponds to a Riemannian metric (Zhang et al., 2019; Luo et al., 2021; Han et al., 2021). Though $\mathrm{SPD}(g_0)$ can be equipped with various different Riemannian metrics, such the log-Euclidean metric and the affine-invariant metric, not all distance functions can be extended to $\mathrm{PSD}(g_0)$ due to the presence of singularities at the boundary points. We define the *tropical Torelli–Wasserstein (TTW) kernel* as follows:

$$k_{\mathrm{TTW}}(G_1, G_2) = \exp(-\gamma W^2(Q(G_1), Q(G_2))).$$

We make the important remark here that the TTW kernel is not positive definite, since the 2-Wasserstein distance is not conditionally negative definite. Effective learning methods have been developed for this setting; see Appendix B.1 for further details. As we will see next, in practice, we do not observe significant instability due to the indefinite nature of TTW.

As a consequence of Theorem 1, the TTE kernel and TTW kernels are invariant to graph refinement, thus, they are well-defined metric graph kernels. We summarize the computation of the TTE kernel and the TTW kernel in Algorithm 2.

---

**Algorithm 2:** Computing the kernel matrix

**Input:** list of $N$ weighted graph $G$, genus bound $g_0$
**Output:** kernel matrix $K$

1 **for** $i \leftarrow 1$ **to** $N$ **do**
2     compute the tropical Torelli matrix $Q_i$ for $G_i$
3     **if** $\dim(Q_i) < g_0$ **then**
4         $Q_i \leftarrow \mathrm{diag}\{Q_i, 0\}$
5     **else**
6         $Q_i \leftarrow Q_i[\mathrm{rand}(g_0), \mathrm{rand}(g_0)]$
7     **end**
8 **end**
9 $K \leftarrow \exp(-\gamma \cdot \mathrm{pairwise\_dist}^2([Q_i]))$

---

### 4.2 COMPUTATIONAL COMPLEXITY

For a list of $N$ matrices $Q_i$ of dimension $g_0$, the time complexity of computing the full kernel matrix is $O(N^2 g_0^2)$ for the TTE kernel and $O(N^2 g_0^3)$ for the TTW kernel if we use eigenvalue decomposition to compute the square root of a matrix. In practice, we find the computation of the TTE kernel matrix is more efficient for sparse graphs since the matrices $Q_i$ can be stored in sparse form which has less memory cost than $O(g_0^2)$.

## 5 EXPERIMENTS

We evaluate our tropical graph kernels with with three experimental studies: (i) **Simulations**: We simulate graphs of different sparsity and test the practical computational time. (ii) **Comparisons**: We compare the performance of our kernels with classical label-free graph kernels on standard benchmark datasets. (iii) **Real data application**: We construct datasets of local urban road networks and demonstrate the applicability of our methods on classification tasks.

**Implementation and Computing Infrastructure.** We use the open source Python library `GraKeL` (Siglidis et al., 2020), which includes all the graph kernels used in in Section 5.2. Unless otherwise specified, we always use the default hyperparameters provided by `GraKeL`. All our experiments are performed on a shared server running with 8 CPUs (Intel Xeon Platinum 8358 (Ice Lake) 2.60GHz), 128 GB of RAM, and a walltime limit of 8 hours.

## 5.1 VERIFYING COMPUTATION TIME ON SYNTHETIC DATASETS

Given node number $n$ and genus $g$, we first generated a line graph with $n$ nodes, and then randomly connected two nodes until the edge number achieves $m = g + n - 1$. We set the range of $n$ from 50 to 140. For each $n$, we generated $N = 10$ synthetic graphs corresponding to each of the following settings of sparsity: (i) **Sparse graphs** such that $g = c_s$ for $c_s = 5, 10, 20$; (ii) **Semi-sparse graphs** such that $g = \lfloor c_{ss}n \rfloor$ for $c_{ss} = 1.0, 1.5, 2.0$; and (iii) **Dense graphs** such that $g = \lfloor n^{1+c_d} \rfloor$ for $c_d = 0.1, 0.2, 0.3$. On each synthetic dataset, we repeated the computation of kernel matrices 10 times. Figure 6 in Appendix B.2 shows how the computation time of the TTE and TTW kernels scales with the number of nodes $n$ across datasets with varying sparsity levels. We observe that the practical computation time for the TTE kernel closely follows the growth pattern of the graph genus, whereas the TTW kernel shows faster growth for denser graphs. Moreover, we see that the computation time for both kernels remains nearly constant on sparse datasets regardless of the change of number of nodes, which highlights their potential for applications involving large sparse graphs.

## 5.2 COMPARING TO EXISTING KERNELS: GRAPH CLASSIFICATION ON BENCHMARKS

We now compare the performance of our graph kernels with existing graph kernels on benchmark datasets for graph classification tasks. Since ours are the first graph kernels designed for metric graphs, there are no prior baselines available in this setting; thus, we compare against classical label-free graph kernels, which represent the most relevant existing alternatives.

**Label-Free Graph Kernels.** Given that metric graph kernels operate on weighted unlabeled graphs, we focus on classical graph kernels that are likewise independent of node/edge labels and attributes; note that this therefore excludes from our comparison set the popular Weisfeiler–Lehman graph kernel (Shervashidze et al., 2011), which depends on node attributes. In particular, we select kernels which can capture global graph structure and are naturally suited to weighted settings. For certain kernels which can be applied to both labeled or unlabeled graphs, we set constant node/edge labels to provide a fair comparison, ensuring that observed performance differences arise from differences in the underlying graph geometry and topology. We compare our kernels with the following five kernels: Edge Histogram (EH) kernel (Sugiyama & Borgwardt, 2015); the Graphlet Sampling (GS) kernel (Shervashidze et al., 2009); the ODD-STh (OS) kernel (Da San Martino et al., 2012); the Shortest Path (SP) kernel (Borgwardt & Kriegel, 2005); the $k$-Core Decomposition ($k$CD) Kernel (Nikolentzos et al., 2018). A detailed description of these kernels and comparison their computational complexity are given in Appendix B.3.

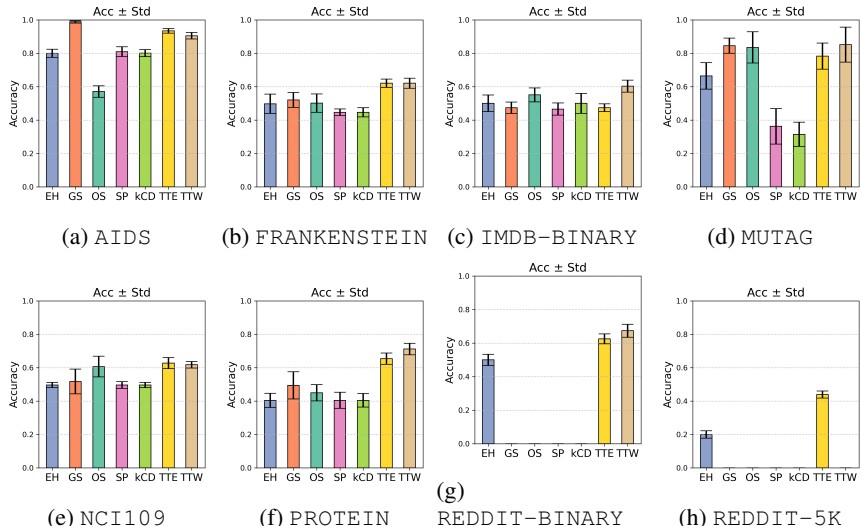

(a) AIDS    (b) FRANKENSTEIN    (c) IMDB-BINARY    (d) MUTAG

(e) NCI109    (f) PROTEIN    (g) REDDIT-BINARY    (h) REDDIT-5K

Figure 2: Classification accuracy of 10-fold cross-validation on benchmark datasets (partial).

**Benchmark Graph Datasets.** We compare the performance of graph kernels on 23 benchmark graph datasets taken from TUDataset (Morris et al., 2020). To adapt to the setting of metric graphs, we

initialize the node/edge labels and weights of graphs by replacing the original node/edge labels by constants. For unweighted graphs, we add random weights to edges from the uniform distribution $\mathrm{Unif}(0, 1)$. To compare computational time, we classify the sparsity of benchmark datasets as follows: Let $\bar{n}$ and $\bar{g}$ be the average number of nodes and genus of a dataset. We identify a dataset as (i) sparse, if $\log(\bar{g}/\bar{n}) < 0$; (ii) semi-sparse, if $0 \leq \log(\bar{g}/\bar{n}) < 1$; (iii) dense, if $\log(\bar{g}/\bar{n}) \geq 1$. A complete description of dataset statistics can be found in Table 5 in Appendix C.1.

**Test Results.** We performed classification using the C-SVM solver in `sklearn` to test the graph kernels. We used the default hyperparameters in `sklearn` and evaluated the SVM model through a 10-fold cross-validation. Figure 2 presents the classification accuracy on 8 datasets; a complete list of classification accuracy can be found in Table 6 in Appendix C.2. We see that the TTE and TTW kernels outperform other graph kernels on most benchmark datasets. Since we excluded the label information from the graphs, the classification accuracy of some classical graph kernels are lower than the results reported in previous studies (Borgwardt et al., 2020; Kriege et al., 2020).

Table 1: Computation time for benchmark datasets (partial). The EH kernel is excluded.

| Name | $\log(\bar{g}/\bar{n})$ | Sparsity | GS | OS | SP | $k$CD | TTE | TTW |
|---|---|---|---|---|---|---|---|---|
| AIDS | -2.26 | S | 9.59 | 44.30 | 106.51 | 231.80 | **2.53** | 175.45 |
| BZR | -2.29 | S | 8.57 | 6.49 | 3.63 | 11.81 | **0.73** | 9.70 |
| MSRC-9 | 0.36 | SS | 111.21 | 5.99 | 2.78 | 12.86 | **1.85** | 31.72 |
| MSRC-21 | 0.45 | SS | 1004.74 | 109.10 | 68.17 | 257.27 | **25.97** | 1497.16 |
| BZR-MD | 2.26 | D | 2328.25 | 0.97 | **0.91** | 63.16 | 19.21 | 1410.91 |
| ER-MD | 2.31 | D | 5503.19 | **1.48** | 2.78 | 119.88 | 31.80 | 5681.93 |

We also tested the computation time of feature maps (`.fit` function in GraKeL) and full kernel matrices (`.transform` function in GraKeL). We excluded the EH kernel since it is the simplest and fastest. Table 1 presents the total computational time of 6 datasets; complete results can be found in Table 7 and 8 in Appendix C.2. We see that the TTE and TTW kernel are particularly efficient on sparse and semi-sparse graph datasets. For the computation of full kernel matrices, TTE is the most efficient method and gives results on all 23 datasets.

## 5.3 URBAN ROAD NETWORK CLASSIFICATION

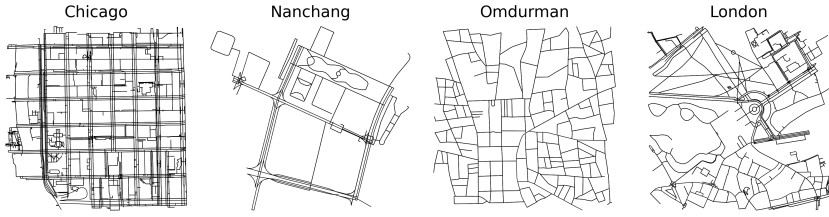

Figure 3: Illustration of URN patterns. From left to right, URNs are taken from Chicago (gridiron), Nanchang (linear), Omdurman (chaotic), and London (tributary).

Urban road networks (URNs) are naturally modeled as metric graphs, where a combinatorial model is built on top of landmarks (nodes) and roads (edges). These networks encode both geometric and topological information, capturing the structural organization of cities. Global URNs present dominant patterns across multiple scales, which are closely related to continents and regions. Based on the study of (Chen et al., 2024), we focus on URN patterns at small scale ($< 500$ meters), grouped into 4 categories: gridiron, linear, chaotic, tributary. Figure 3 presents sample cities from each category.

**Constructing Datasets.** The spatial data of URNs were retrieved from the OpenStreetMap (OSM) database (OpenStreetMap contributors, 2017), queried by the open source Python library OSMnx (Boeing, 2025). We constructed 8 datasets for classification: First, we picked a city for each category. For a city with latitude–longitude coordinate $C = (x, y)$, we retrieved a large road network centered at $C$ with global radius $R$ and then randomly sampled $L$ landmarks from the large network. For each

landmark, we retrieved a small road network with local radius $r$. See Table 9 in Appendix D.1 for full details.

**Classification Results.** We performed classification using C-SVM for 4 graph kernels: OS, SP, TTE, and TTW. Figure 4 displays an exploratory dimension reduction analysis via kernel PCA. Table 2 shows the complete list of dataset statistics, classification accuracy and computational time. The TTE and TTW kernels outperform the other graph kernels on most datasets. Notably, URNs are typically sparse, which makes the TTE and TTW kernels particularly well-suited for these datasets, where other kernels fail due to memory limitations. Additional results can be found in Appendix D.2.

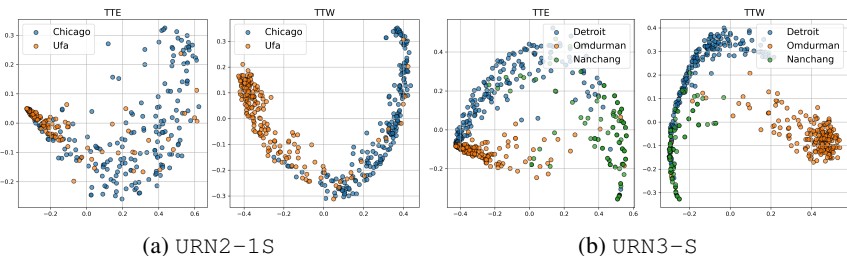

(a) `URN2-1S`                    (b) `URN3-S`

Figure 4: Dimension reduction by kernel PCA. The low dimensional representations indicate that TTE and TTW kernels are able classify different URNs.

Table 2: Classification results for URN datasets. In the table, $\bar{n}, \bar{g}$ are the average numbers of nodes and genus. The accuracy is measured in percentage and time is measured in seconds. "M" indicates out of memory. The maximal memory is set to 128GB.

|  |  |  | OS | | SP | | TTE | | TTW | |
|---|---|---|---|---|---|---|---|---|---|---|
| Name | $\bar{n}$ (*std.*) | $\bar{g}$ (*std.*) | Acc (*std.*) | Time | Acc (*std.*) | Time | Acc (*std.*) | Time | Acc (*std.*) | Time |
| 2-1S | 120.40 (*66.73*) | 54.26 (*34.47*) | 84.00 (*12.21*) | 188.70 | 92.25 (*5.30*) | 21.01 | 87.50 (*5.24*) | 14.96 | **93.50** (*4.90*) | 114.63 |
| 2-1M | 362.35 (*203.81*) | 187.96 (*123.03*) | M | M | M | M | 67.10 (*7.48*) | 105.43 | **89.75** (*3.61*) | 1357.85 |
| 2-2S | 96.40 (*77.20*) | 38.84 (*36.89*) | 90.50 (*1.98*) | 252.25 | **94.67** (*3.23*) | 39.06 | 91.50 (*3.11*) | 17.82 | 92.67 (*3.43*) | 166.45 |
| 2-2M | 289.60 (*239.37*) | 123.00 (*121.43*) | M | M | M | M | 92.67 (*3.82*) | 119.28 | **94.50** (*2.99*) | 1553.34 |
| 3-S | 55.02 (*51.09*) | 22.31 (*24.06*) | 62.50 (*15.69*) | 99.02 | 84.01 (*3.43*) | 13.27 | 87.67 (*4.03*) | 7.25 | **93.17** (*2.83*) | 79.36 |
| 3-M | 250.49 (*206.92*) | 131.61 (*128.22*) | M | M | M | M | 75.60 (*2.80*) | 143.54 | **84.13** (*2.63*) | 2750.13 |
| 4-S | 81.82 (*74.89*) | 37.68 (*38.39*) | **84.38** (*3.80*) | 365.67 | 83.00 (*4.58*) | 67.85 | 81.12 (*4.12*) | 22.07 | 81.00 (*4.36*) | 287.45 |
| 4-M | 283.36 (*238.34*) | 133.03 (*125.64*) | M | M | M | M | 68.12 (*3.80*) | 155.03 | **77.25** (*3.70*) | 3187.99 |

## 6 DISCUSSION

We proposed new graph kernels which are extendable to the space of metric graphs. We adapted the classical Torelli map from tropical geometry to weighted graphs, embedding them into the space of PSD matrices. We constructed our kernels using two distance functions. Through comprehensive experiments, we demonstrated the effectiveness and applicability of our metric graph kernels.

**Limitations and Future Work.** Graphs that are trees have genus 0, so the tropical Torelli map is trivial. Consequently, the resulting kernels identify all trees. Also our method does not incorporate label or attribute information. Developing a principled way to define and integrate labels on metric graphs remains an open challenge and is a promising direction for future work.

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

## A    FURTHER DETAILS OF THE TROPICAL TORELLI MAP

In this section of the Appendix, we provide further background on the tropical geometry of the tropical Torelli map; the proof of our main Theorem 1; and an explicit example on how to compute our SPD matrix $Q$.

### A.1    GEOMETRIC BACKGROUND

In classical algebraic geometry over the complex numbers, one of the fundamental results is that every smooth projective algebraic curve $X$ of genus $g$ encodes rich geometric information in a special complex torus $\mathbb{C}^g/L$. This object is known as the Jacobian variety of the curve $\mathrm{Jac}(X)$. The classical Torelli map is a natural and powerful construction that assigns to each curve $X$ its Jacobian $\mathrm{Jac}(X)$. This map is injective, meaning that the Jacobian retains enough information to uniquely recover the curve. The Torelli map thus plays a central role in the study of algebraic curves by linking their intrinsic geometry to the rich structure of abelian varieties (Miranda, 1995).

In tropical geometry—a piecewise-linear analog of algebraic geometry—a tropical curve is represented not by a smooth curve, but by a metric graph $\Gamma$. Just as classical algebraic curves have associated Jacobian varieties, tropical curves have the tropical Jacobian variety, which is a real torus of the form $\mathrm{Jac}(\Gamma) = \mathbb{R}^g/L$. The tropical Torelli map is the function that assigns to each tropical curve $\Gamma$ to its corresponding tropical Jacobian $\mathrm{Jac}(\Gamma)$. This map captures the essential geometric and topological information of the graph in a purely combinatorial and metric framework (Brannetti et al., 2011).

In computations we have to choose a vector space basis for $\mathbb{R}^n$ and a lattice basis for $L$, hence the tropical Jacobian can be represented in the following equivalent forms:

(I) The tropical form $(\mathbb{R}^g/Q, Q^{-1})$, where the lattice is spanned by the column vectors of $Q$, and the inner product on $\mathbb{R}^g$ is represented by $Q^{-1}$. Under this form, the squared distance function is given by

$$d^2_{(\mathrm{I})}([x],[y]) = \min_{w \in \mathbb{Z}^g} \left( (x - y - Qw)^\top Q^{-1}(x - y - Qw) \right).$$

Form (I) is particularly useful for some computations in tropical geometry such as the tropical Abel–Jacobi transform of metric graphs (Cao & Monod, 2025).

(II) The standard lattice form $(\mathbb{R}^g/\mathbb{Z}^g, Q)$, where the lattice is $\mathbb{Z}^n$, and the inner product on $\mathbb{R}^g$ is represented by $Q$. Under this form, the squared distance function is given by

$$d^2_{(\mathrm{II})}([x],[y]) = \min_{w \in \mathbb{Z}^g} \left( (x - y - w)^\top Q(x - y - w) \right).$$

Our main algorithm in Section 3.2 of the tropical Torelli map for weighted graphs is based on form (II).

(III) The standard inner product form $(\mathbb{R}^g/\sqrt{Q}, \|\cdot\|_2)$ where the lattice is spanned by the column vectors of $\sqrt{Q}$, and the inner product on $\mathbb{R}^g$ is the standard $L_2$ inner product. Under this form, the squared distance function is given by

$$d^2_{(\mathrm{III})}([x],[y]) = \min_{w \in \mathbb{Z}^g} \left\| x - y - \sqrt{Q}w \right\|_2^2.$$

Form (III) is the motivation for us to consider the Bures–Wasserstein distance in the construction of tropical Torelli–Wasserstein graph kernel in Section 4.1.

## A.2 PROOF OF THEOREM 1

We first show that if $G$ has a generic length function, then $G$ can be oriented in a canonical way through the following operations: (1) for any $e \in T$ with endpoints $u, v$, define $\omega(u)$ as the length of the other tree edge incident to $u$. We set $\omega(u) = 0$ if no such edge exists; if there is more than one incident edge, we choose $\omega(u)$ to be the minimum length among all incident tree edges. Then orient $e$ from low $\omega$ value to the high $\omega$ value; (2) for any $e \notin T$ with endpoints $u, v$, define $w(u)$ as the length of the unique tree edge incident to $u$ that lies on the path from $u$ to $v$ in $T$. Then orient $e$ from low $\omega$ value to the high $\omega$ value.

In Algorithm 1, the edges not in the minimal spanning $T$ are sorted by length. With a generic length function $\ell$, the minimal spanning tree $T$ is unique and the edge lengths are distinct, thus the 1-cycles $\sigma_1, \ldots, \sigma_g$ are uniquely determined. It suffices to show that the cycle-edge matrix is independent of the ordering of edges in $T$. In fact, suppose $e'_1, \ldots, e'_{n-1}$ is another ordering of edges in $T$. Then there exists a permutation matrix $P \in S_{n-1}$ such that $M'_T = M_T P$ and $L'_T = P^\top L_T P$. Therefore,

$$Q' = M'_T L'_T M'^{\top}_T + L_g = M_T P P^\top L_T P P^\top M_T^\top + L_g = Q.$$

Then we show that the matrix $Q$ is invariant under edge subdivision. Without loss of generality we assume that $e_s$ is subdivided into $e'_s$ and $e'_{s+1}$. For remaining edges we set $e'_k = e_k$ if $k < s$, and $e'_k = e_{k-1}$ if $k > s + 1$. For cycle-edge incidence matrices we have

$$\begin{cases} M'_{ik} = M_{ik}, \ 1 \le i \le g \, , 1 \le k \le s, \\ M'_{ik} = M_{i(k-1)}, \ 1 \le i \le g, \ s + 2 \le k \le m + 1, \\ M'_{i(s+1)} = M_{is}, \ 1 \le i \le g. \end{cases}$$

For edge-length matrices we have

$$\begin{cases} L'_{kk} = L_{kk}, \ 1 \le k < s, \\ L'_{kk} = L_{(k-1)(k-1)}, \ s + 1 < k \le m + 1, \\ L'_{ss} + L'_{(s+1)(s+1)} = L_{ss}. \end{cases}$$

Therefore, for any $1 \le i, j \le g$, we have

$$\begin{aligned} Q'_{ij} &= \sum_{k=1}^{m+1} M'_{ik} L'_{kk} M'_{jk} \\ &= \sum_{k=1}^{s-1} M_{ik} L_{kk} M_{jk} + M'_{is} L'_{ss} M'_{js} + M'_{i(s+1)} L'_{(s+1)(s+1)} M'_{j(s+1)} + \sum_{k=s+2}^{m+1} M'_{ik} L'_{kk} M'_{jk} \\ &= \sum_{k=1}^{s-1} M_{ik} L_{kk} M_{jk} + M_{is}(L'_{(s+1)(s+1)} + L'_{ss}) M_{js} + \sum_{k=s+1}^{m} M_{ik} L_{kk} M_{jk} \\ &= \sum_{k=1}^{m} M_{ik} L_{kk} M_{jk} = Q_{ij}. \end{aligned}$$

## A.3 EXAMPLE

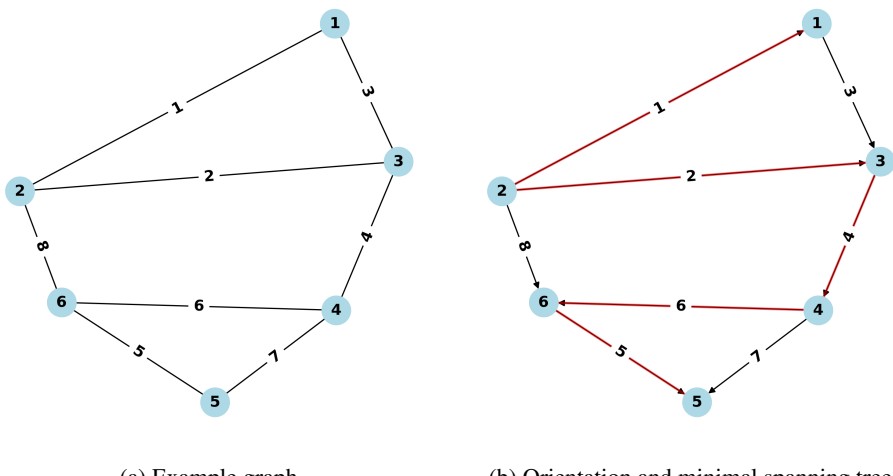

(a) Example graph     (b) Orientation and minimal spanning tree

Figure 5: Illustration of Algorithm 1. On the right panel, the orientation is indicated by arrow and the minimal spanning tree is colored in red.

We demonstrate the computation of the tropical Torelli map on a toy example shown in Figure 5a. The graph $G$ has 6 nodes and 8 edges. The length of each edge is defined as $\ell(e_i) = i$. We compute the minimal spanning tree as $T = \{e_1, e_2, e_4, e_5, e_6\}$, which is shown in Figure 5b. After fixing an orientation, each edge not in $T$ determines a 1-cycle in the following:

$$\sigma_1 = e_3 - e_2 + e_1,$$
$$\sigma_2 = e_7 - e_5 - e_6,$$
$$\sigma_3 = e_8 - e_6 - e_4 - e_2.$$

Thus the cycle-edge incidence matrix is given by

$$M = \begin{bmatrix} 1 & -1 & 1 & 0 & 0 & 0 & 0 & 0 \\ 0 & 0 & 0 & 0 & -1 & -1 & 1 & 0 \\ 0 & -1 & 0 & -1 & 0 & -1 & 0 & 1 \end{bmatrix}.$$

The matrix of the tropical Torelli map can be computed by

$$Q = MLM^\top = \begin{bmatrix} 6 & 0 & 2 \\ 0 & 18 & 6 \\ 2 & 6 & 20 \end{bmatrix}.$$

## B GRAPH KERNELS

In this section of the Appendix, we provide further technical background on graph kernels and their construction, and specifically discuss the case of indefinite kernels. We also provide numerical verifications of our developed theory. Finally, we give full details on the graph kernels used for comparison in our benchmarking studies.

### B.1 BACKGROUND ON KERNELS

Kernels form a foundational class of similarity functions in machine learning and data analysis, which are often used to capture nonlinear relationships by embedding data into high-dimensional feature spaces. Let $\mathcal{X}$ be a set and $k : \mathcal{X} \times \mathcal{X} \to \mathbb{R}$ be a symmetric function. Commonly $k$ is required to be positive definite in the sense that for any finite set of points $x_1, x_2, \ldots, x_n \in \mathcal{X}$ the matrix $K_{ij} = k(x_i, x_j)$ is positive semi-definite, i.e., for any real coefficients $c_1, c_2, \ldots, c_n$, we have

$$\sum_{i=1}^{n}\sum_{j=1}^{n} c_i c_j k(x_i, x_j) \geq 0.$$

For any positive definite kernel $k$, there exists a unique Hilbert space $\mathcal{H}$ known as the reproducing kernel Hilbert space (RKHS), along with a map $\phi : \mathcal{X} \to \mathcal{H}$ called the feature map, such that the value of the kernel is given by the inner product of the feature map (Wainwright, 2019)

$$k(x, y) = \langle \phi(x), \phi(y) \rangle_{\mathcal{H}}.$$

A typical construction of kernel is through the distance on a space. Let $d(x, y)$ be a distance function on $\mathcal{X}$, the distance is said to be conditionally negative type if for any finite set of points $x_1, x_2, \ldots, x_n \in \mathcal{X}$ and any real coefficients $c_1, c_2, \ldots, c_n$ such that $\sum_{i=1}^{n} c_i = 0$, we have

$$\sum_{i=1}^{n}\sum_{j=1}^{n} c_i c_j d(x_i, x_j) \leq 0.$$

If $d(x, y)$ is conditionally negative type, then the kernel defined by $k(x, y) = \exp(-\gamma d(x, y)^2)$ is positive definite for any $\gamma > 0$, which is known as Schoenberg's theorem (Sejdinovic et al., 2013). The Euclidean distance is a typical example of conditionally negative type distance. However, the 2-Wasserstein distance is not of conditionally negative type (Peyré et al., 2019), and thus the induced kernel is indefinite. In practice indefinite kernels are also widely used, and the corresponding learning methodology is called indefinite learning. In general, an indefinite kernel corresponds to a Reproducing Kernel Kreĭn Space (RKKS), on which many RKHS-based techniques can be extended (Oglic & Gärtner, 2018; Togninalli et al., 2019).

### B.2 NUMERICAL VERIFICATIONS

**Subsampling Stability.**

We provide experimental verifications of our subsampling method described in Section 4 to map matrices to the same dimension by extracting a submatrix by random subsampling.

We carried out additional experiments and display partial results for tropical Torelli matrix projections to different dimensions. The experimental setting is the following: For a set of graphs with genus $g$=200, we project the tropical Torelli matrices to spaces of dimension $g_0$, ranging from 50 to 150. Then we compute the relative $L_2$ error between the original matrix and the projected matrix. The mean error and standard deviation are displayed in the following table.

Table 3: Numerical Stability of Random Principal Submatrix Selection. The average relative $L_2$ errors and standard deviations are shown with respect to an increasing projection dimension $g_0 = 50 \sim 150$.

| TTW | 0.712 | 0.667 | 0.608 | 0.565 | 0.508 | 0.465 | 0.426 |
|-----|-------|-------|-------|-------|-------|-------|-------|
|     | *(0.002)* | *(0.002)* | *(0.005)* | *(0.002)* | *(0.006)* | *(0.004)* | *(0.004)* |
| TTE | 0.914 | 0.882 | 0.844 | 0.799 | 0.763 | 0.727 | 0.667 |
|     | *(0.010)* | *(0.019)* | *(0.006)* | *(0.004)* | *(0.027)* | *(0.016)* | *(0.011)* |

From these results, we see that the relative error changes slowly with respect to dimension.

**Verifying Computation Time.**

We ran experiments on synthetic datasets to verify the computation runtime on our proposed classes of sparsity based on genus. The plots below show that the runtime to compute the TTE kernel is in alignment with the growth of the graph genus, while the runtime to compute the TTW kernel grows faster with graph density.

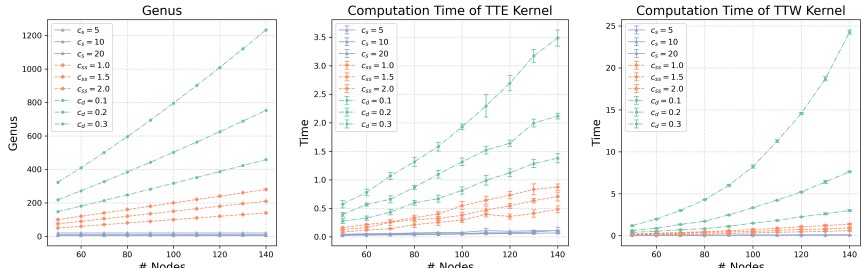

Figure 6: Verifying computation time. From left to right: the first figure plots genus as a function of number of nodes; the second and third figures show the computation time of kernel matrices.

### B.3    COMPARED GRAPH KERNELS

We list the graph kernels used for comparison in the benchmark dataset experiments below:

- **The Edge Histogram (EH) kernel**. The EH kernel compares graphs based on the histogram of discrete edge weights or labels (Sugiyama & Borgwardt, 2015). It is the most computationally efficient graph kernel. However it cannot capture the graph topology.

- **The Graphlet Sampling (GS) kernel**. The GS kernel measures the similarity between graphs via counting matching graphlets, i.e., small subgraphs with $k$ vertices (Shervashidze et al., 2009). Since enumerating all graphlets is computationally expensive, we set the maximal size of a graphlet to be $k = 6$, and the maximal number of samples to be 300.

- **The ODD-STh (OS) kernel** The OS kernel compares graphs based on the Directed Acyclic Graph (DAG) decomposition (Da San Martino et al., 2012). For each node in a graph, a breadth-first search (BFS) tree or DAG is constructed up to a fixed depth, capturing the local neighborhood structure in a rooted format. For unlabeled (or constantly labeled) graphs, the OS kernel is able to capture the structural differences.

- **The Shortest Path (SP) kernel**. The SP kernel compares graphs based on the shortest path lengths between all pairs of nodes (Borgwardt & Kriegel, 2005). The original SP kernel uses a base kernel function on the node labels at the path endpoints and the path lengths, which is computationally expensive. We use the simplified version using the histogram of shortest paths, which is also the implemented version in `GraKeL`.

- **The $k$-Core Decomposition ($k$CD) kernel**. The $k$CD kernel compares graphs based on the $k$-core decomposition of a graph (Nikolentzos et al., 2018). The $k$-core of a graph is the maximal subgraph in which all nodes have degree at least $k$. This decomposition assigns a core number to each node, indicating its topological depth or centrality within the graph. The $k$CD kernel uses the distribution of these core numbers as the basis for comparison between graphs.

Table 4: Computational complexity for different graph kernels. In the table, `.fit` time is the time complexity to compute the features for all graphs in a dataset; `.transform` time is the time complexity to compute the full kernel matrix. $N$ is the number of graphs in a dataset.

| Name | Parameters | `.fit` Time | `.transform` Time |
|---|---|---|---|
| EH | $m$: # edges | $O(N \cdot m)$ | $O(N^2 \cdot m)$ |
| GS | $B$: # graphlet samples; $G_k$: # $k$-graphlets | $O(N \cdot B)$ | $O(N^2 \cdot G_k)$ |
| OS | $n$: # nodes; $d$: node degree; $h$: subtree depth; $T_h$: # tree types of depth $h$ | $O(N \cdot n \cdot d^h)$ | $O(N^2 \cdot T_h)$ |
| SP | $n$: # nodes | $O(N \cdot n^2 \log n)$ | $O(N^2 \cdot n^2)$ (histogram) $O(N^2 \cdot n^4)$ (all pairs) |
| $k$CD | $n$: # nodes; $m$: # edges; $C_k$: max core number | $O(N \cdot (n + m))$ | $O(N^2 \cdot C_k)$ |
| TTE | $n$: # nodes; $g$: genus | $O(N \cdot g^2 n)$ | $O(N^2 \cdot g^2)$ |
| TTW | $n$: # nodes; $g$: genus | $O(N \cdot g^2 n)$ | $O(N^2 \cdot g^3)$ |

# C ADDITIONAL EXPERIMENTAL RESULTS ON BENCHMARK DATASETS

## C.1 DATASET STATISTICS

Table 5: Benchmark dataset statistics. In the table, $N$ is the number of graphs in a dataset; $\bar{n}, \bar{m}, \bar{g}$ are the average numbers of nodes, edges, and genus. In the sparsity column, "S", "SS" and "D" are short for "Sparse", "Semi-Sparse" and "Dense".

| Name | $N$ | $\bar{n}$ (std.) | $\bar{m}$ (std.) | $\bar{g}$ (std.) | $\log(\bar{g}/\bar{n})$ | Sparsity |
|---|---|---|---|---|---|---|
| AIDS | 2000 | 15.59 (13.59) | 16.20 (15.01) | 1.63 (1.67) | -2.26 | S |
| BZR | 405 | 35.75 (7.26) | 38.36 (7.69) | 3.61 (0.87) | -2.29 | S |
| BZR-MD | 306 | 21.30 (4.19) | 225.06 (85.83) | 204.75 (81.70) | 2.26 | D |
| COIL-DEL | 3900 | 21.54 (13.22) | 54.24 (38.47) | 33.70 (25.27) | 0.45 | SS |
| COX2 | 467 | 41.22 (4.03) | 43.45 (4.27) | 3.22 (0.42) | -2.55 | S |
| COX2-MD | 303 | 26.28 (2.46) | 335.12 (65.19) | 309.84 (62.75) | 2.47 | D |
| DD | 1178 | 284.32 (272.00) | 715.66 (693.91) | 432.36 (426.25) | 0.42 | SS |
| DHFR | 756 | 42.43 (9.06) | 44.54 (9.25) | 3.12 (0.65) | -2.61 | S |
| DHFR-MD | 393 | 23.87 (4.50) | 283.02 (106.85) | 260.15 (102.40) | 2.39 | D |
| ENZYMES | 600 | 32.46 (14.87) | 62.14 (25.50) | 30.75 (13.01) | -0.05 | S |
| ER-MD | 446 | 21.33 (6.01) | 234.85 (133.47) | 214.52 (127.63) | 2.31 | D |
| FRANKEN -STEIN | 4337 | 16.83 (10.45) | 17.88 (11.60) | 2.07 (1.67) | -2.10 | S |
| IMDB -BINARY | 1000 | 19.77 (10.06) | 96.53 (105.60) | 77.76 (98.22) | 1.37 | D |
| IMDB -MULTI | 1500 | 13.00 (8.52) | 65.94 (110.78) | 53.93 (103.84) | 1.42 | D |
| MSRC-9 | 221 | 40.58 (5.27) | 97.94 (15.14) | 58.36 (10.18) | 0.36 | SS |
| MSRC-21 | 563 | 77.52 (12.27) | 198.32 (36.81) | 121.80 (25.20) | 0.45 | SS |
| MSRC-21C | 209 | 40.28 (5.82) | 96.60 (16.44) | 57.33 (10.86) | 0.35 | SS |
| MUTAG | 188 | 17.93 (4.58) | 19.79 (5.68) | 2.86 (1.30) | -1.84 | S |
| NCI1 | 4110 | 29.76 (13.55) | 32.30 (14.93) | 3.62 (1.98) | -2.11 | S |
| NCI109 | 4127 | 29.57 (13.55) | 32.13 (14.96) | 3.64 (1.98) | -2.09 | S |
| PROTEINS | 1113 | 39.05 (45.76) | 72.82 (84.60) | 34.84 (40.56) | -0.11 | S |
| REDDIT -BINARY | 2000 | 429.62 (554.05) | 497.75 (622.99) | 70.61 (85.56) | -1.81 | S |
| REDDIT -MULTI-5K | 4999 | 508.51 (452.57) | 594.87 (566.77) | 90.08 (133.61) | -1.73 | S |

## C.2 FULL TEST RESULTS

Table 6: Classification accuracy of 10-fold cross-validation. In the table, the accuracy is measured by percentage (%); "M" stands for out of memory. The max memory is set to be 128GB.

| Name | EH (std.) | GS (std.) | OS (std.) | SP (std.) | $k$CD (std.) | TTE (std.) | TTW (std.) |
|---|---|---|---|---|---|---|---|
| AIDS | 80.00 (2.44) | **98.80** (0.68) | 57.10 (37.47) | 81.00 (2.95) | 80.10 (2.07) | 93.40 (1.37) | 90.50 (1.94) |
| BZR | 78.78 (5.44) | 57.54 (10.70) | 76.55 (11.18) | 78.74 (3.98) | 78.79 (6.69) | 82.74 (7.67) | **83.71** (4.80) |
| BZR-MD | 48.72 (7.44) | 51.00 (8.82) | **61.53** (12.77) | 48.71 (7.13) | 48.68 (6.05) | 48.13 (8.89) | 51.26 (4.95) |
| COIL-DEL | 1.00 (0.33) | **15.03** (1.65) | 14.72 (2.47) | 1.00 (0.60) | 1.01 (0.51) | 1.97 (0.66) | 3.10 (1.03) |
| COX2 | 78.15 (3.49) | 76.23 (3.85) | 72.62 (8.60) | 78.17 (4.33) | 78.16 (3.26) | 78.17 (5.14) | **78.21** (7.69) |
| COX2-MD | 47.81 (8.51) | 48.85 (6.42) | 51.83 (7.92) | 48.20 (6.53) | 48.54 (6.21) | **52.20** (6.58) | 47.20 (10.68) |
| DD | 41.35 (4.41) | 51.69 (8.87) | M | M | M | **58.66** (3.35) | M |
| DHFR | 60.96 (3.86) | 56.10 (6.61) | **74.85** (4.88) | 61.00 (4.14) | 60.99 (4.55) | 58.86 (6.19) | 60.86 (7.08) |
| DHFR-MD | 67.90 (5.39) | 60.25 (17.33) | 66.13 (7.37) | 67.00 (6.50) | 67.90 (5.46) | 67.94 (7.54) | **67.95** (7.40) |
| ENZYMES | 16.00 (2.60) | 21.67 (6.45) | **22.00** (6.74) | 16.67 (6.32) | 16.83 (5.40) | 12.67 (3.00) | 18.67 (6.05) |
| ER-MD | 49.41 (6.96) | 46.23 (10.77) | 58.55 (11.63) | 59.41 (9.95) | 59.37 (8.55) | 59.40 (5.18) | **59.43** (6.58) |
| FRANKEN-STEIN | 49.76 (5.78) | 52.04 (4.43) | 50.15 (5.46) | 44.64 (2.02) | 44.64 (2.74) | **62.09** (2.48) | 62.07 (2.99) |
| IMDB-BINARY | 50.00 (4.98) | 47.40 (3.38) | 55.10 (4.16) | 46.60 (3.61) | 50.00 (6.00) | 47.40 (2.33) | **60.30** (3.58) |
| IMDB-MULTI | 33.30 (4.33) | 34.87 (3.53) | 34.53 (2.93) | 33.33 (3.20) | 33.33 (4.39) | 31.00 (3.42) | **40.40** (2.35) |
| MSRC-9 | 7.23 (4.15) | **20.34** (7.30) | 14.9 (9.47) | 9.51 (4.29) | 10.00 (5.68) | 5.85 (3.42) | 5.40 (3.84) |
| MSRC-21 | 3.00 (2.49) | **9.95** (4.07) | 8.52 (3.16) | 4.09 (2.27) | 5.33 (1.77) | 4.09 (1.80) | 4.61 (1.17) |
| MSRC-21C | 8.67 (5.33) | **18.14** (8.42) | 13.38 (4.63) | 10.00 (5.81) | 8.57 (7.00) | 11.45 (8.81) | 10.55 (6.36) |
| MUTAG | 66.46 (7.92) | 84.53 (4.55) | 83.51 (9.38) | 36.26 (10.65) | 31.46 (7.23) | 78.30 (7.84) | **85.18** (10.43) |
| NCI1 | 49.95 (1.85) | 57.71 (7.93) | 56.11 (9.56) | 49.95 (1.77) | 49.95 (2.91) | 61.85 (1.96) | **62.06** (2.03) |
| NCI109 | 49.63 (1.56) | 51.74 (7.44) | 60.67 (6.15) | 49.62 (2.03) | 49.62 (1.58) | **62.73** (3.29) | 61.76 (1.99) |
| PROTEINS | 40.43 (4.23) | 49.41 (8.10) | 44.93 (4.87) | 40.44 (4.82) | 40.43 (4.08) | 65.41 (3.40) | **71.16** (3.49) |
| REDDIT-BINARY | 50.00 (3.35) | M | M | M | M | 62.50 (2.97) | **67.35** (3.83) |
| REDDIT-MULTI-5K | 20.01 (2.23) | M | M | M | M | **43.89** (2.08) | M |

Table 7: Computation time of features (`.fit` function in `GraKeL`). The wall time is measured in seconds. The edge histogram kernel is excluded.

| Name | Sparsity | GS | OS | SP | $k$CD | TTE | TTW |
|---|---|---|---|---|---|---|---|
| AIDS | S | 9.46 | 19.41 | 4.79 | 11.01 | 2.65 | **1.59** |
| BZR | S | 8.29 | 3.87 | 1.87 | 5.19 | **1.08** | 1.21 |
| BZR-MD | D | 2388.48 | **0.79** | 1.49 | 50.44 | 14.79 | 11.56 |
| COIL-DEL | SS | 1026.84 | 244.50 | **17.03** | 77.47 | 25.78 | 34.13 |
| COX2 | S | 10.37 | 5.09 | 3.71 | 7.90 | 1.50 | **1.20** |
| COX2-MD | D | 9842.24 | **1.35** | 2.50 | 110.06 | 15.42 | 27.31 |
| DD | SS | 8053.55 | M | M | M | **386.69** | M |
| DHFR | S | 17.31 | 11.62 | 6.60 | 17.69 | **1.41** | 2.57 |
| DHFR-MD | D | 4819.32 | **1.43** | 2.73 | 113.35 | 25.23 | 23.94 |
| ENZYMES | S | 60.51 | 11.67 | 6.55 | 15.08 | **2.96** | 6.35 |
| ER-MD | D | 5326.07 | **1.41** | 3.22 | 89.17 | 21.58 | 19.51 |
| FRANKENSTEIN | S | 44.41 | 86.96 | 6.07 | 34.73 | **3.86** | 7.78 |
| IMDB-BINARY | D | 8332.79 | **2.55** | 3.49 | 69.06 | 10.22 | 21.69 |
| IMDB-MULTI | D | 12346.53 | **2.24** | 2.90 | 111.16 | 10.98 | 11.83 |
| MSRC-9 | SS | 110.96 | 3.26 | 3.91 | 9.41 | **2.16** | 2.31 |
| MSRC-21 | SS | 1222.51 | 50.79 | 36.71 | 111.15 | 24.60 | **15.29** |
| MSRC-21C | SS | 104.49 | 3.69 | **1.75** | 8.81 | 3.52 | 3.80 |
| MUTAG | S | 1.31 | 0.38 | **0.30** | 1.10 | 0.89 | 0.96 |
| NCI1 | S | 48.30 | 173.60 | 15.51 | 50.22 | **7.34** | 10.86 |
| NCI109 | S | 57.46 | 174.41 | 31.63 | 50.12 | 7.50 | **7.04** |
| PROTEINS | S | 163.99 | 120.98 | 22.70 | 144.59 | 11.44 | **9.80** |
| REDDIT-BINARY | S | M | M | M | M | **136.88** | 148.57 |
| REDDIT-MULTI-5K | S | M | M | M | M | **374.67** | M |

Table 8: Computation time of full kernel matrices (`.transform` function in `GraKeL`). The wall time is measured in seconds. The edge histogram kernel is excluded.

| Name | Sparsity | GS | OS | SP | $k$CD | TTE | TTW |
|---|---|---|---|---|---|---|---|
| AIDS | S | 9.59 | 44.30 | 106.51 | 231.80 | **2.53** | 175.45 |
| BZR | S | 8.57 | 6.49 | 3.63 | 11.81 | **0.73** | 9.70 |
| BZR-MD | D | 2328.25 | 0.97 | **0.91** | 63.16 | 19.21 | 1410.91 |
| COIL-DEL | SS | 939.85 | 684.63 | 1132.91 | 3631.45 | **30.89** | 5584.44 |
| COX2 | S | 10.59 | 7.66 | 10.20 | 21.16 | **1.29** | 12.96 |
| COX2-MD | D | 9801.29 | 1.52 | **1.18** | 132.11 | 23.93 | 3872.51 |
| DD | SS | 7730.73 | M | M | M | **392.03** | M |
| DHFR | S | 17.27 | 22.72 | 30.14 | 118.15 | **1.24** | 43.51 |
| DHFR-MD | D | 5078.20 | **1.62** | 1.63 | 145.09 | 42.59 | 5970.10 |
| ENZYMES | S | 60.50 | 26.08 | 18.49 | 42.26 | **2.65** | 147.04 |
| ER-MD | D | 5503.19 | **1.48** | 2.78 | 119.88 | 31.80 | 5681.93 |
| FRANKENSTEIN | S | 43.33 | 262.12 | 714.15 | 2198.13 | **3.79** | 1582.95 |
| IMDB-BINARY | D | 8637.97 | **4.19** | 13.49 | 168.42 | 11.68 | 4152.41 |
| IMDB-MULTI | D | 12528.71 | **3.99** | 20.94 | 388.92 | 12.59 | 6400.96 |
| MSRC-9 | SS | 111.21 | 5.99 | 2.78 | 12.86 | **1.85** | 31.72 |
| MSRC-21 | SS | 1004.74 | 109.10 | 68.17 | 257.27 | **25.97** | 1497.16 |
| MSRC-21C | SS | 104.52 | 6.48 | **1.30** | 12.18 | 3.38 | 57.19 |
| MUTAG | S | 1.52 | 0.77 | 0.52 | 1.83 | **0.35** | 2.64 |
| NCI1 | S | 48.56 | 592.90 | 1675.78 | 4458.72 | **7.30** | 934.91 |
| NCI109 | S | 57.46 | 567.12 | 3309.95 | 4490.26 | **7.69** | 994.78 |
| PROTEINS | S | 162.10 | 222.92 | 171.15 | 979.98 | **11.56** | 648.14 |
| REDDIT-BINARY | S | M | M | M | M | **133.08** | 9146.96 |
| REDDIT-MULTI-5K | S | M | M | M | M | **359.45** | M |

# D  ADDITIONAL EXPERIMENTAL RESULTS ON URN DATASETS

## D.1  CONSTRUCTION OF URN DATASETS

Table 9: Summary of URN datasets. In the table, $N$ is the number of graphs in a dataset; $R$ is global radius and $r$ is local radius, both measured in meters.

| Name | $N$ | $R$ | $r$ | Cities (Coordinates) | Patterns |
|---|---|---|---|---|---|
| URN2-1S | 400 | 1000 | 150 | Chicago: $(41.87, -87.63)$
Ufa: $(54.73, 55.95)$ | Gridiron
Chaotic |
| URN2-1M | 400 | 2000 | 300 | Chicago: $(41.87, -87.63)$
Ufa: $(54.73, 55.95)$ | Gridiron
Chaotic |
| URN2-2S | 600 | 1000 | 150 | Wuhan: $(30.52, 114.35)$
London: $(51.50, -0.14)$ | Linear
Tributary |
| URN2-2M | 600 | 2000 | 300 | Wuhan: $(30.52, 114.35)$
London: $(51.50, -0.14)$ | Linear
Tributary |
| URN3-S | 600 | 1000 | 150 | Detroit: $(42.34, -83.05)$
Omdurman: $(15.64, 32.48)$
Nanchang: $(28.68, 115.85)$ | Gridiron
Chaotic
Linear |
| URN3-M | 750 | 2000 | 300 | Seattle: $(47.62, -122.35)$
Ufa: $(54.73, 55.95)$
Tianjin: $(39.12, 117.17)$ | Gridiron
Chaotic
Linear |
| URN4-S | 800 | 1000 | 150 | Chicago: $(41.87, -87.63)$
Omdurman: $(15.64, 32.48)$
Shenyang: $(41.80, 123.43)$
Paris: $(48.87, 2.29)$ | Gridiron
Chaotic
Linear
Tributary |
| URN4-M | 800 | 2000 | 300 | Chicago: $(41.87, -87.63)$
Kano: $(12.01, 8.59)$
Shenyang: $(41.80, 123.43)$
Berlin: $(52.52, 13.40)$ | Gridiron
Chaotic
Linear
Tributary |

## D.2  CONFUSION MATRICES

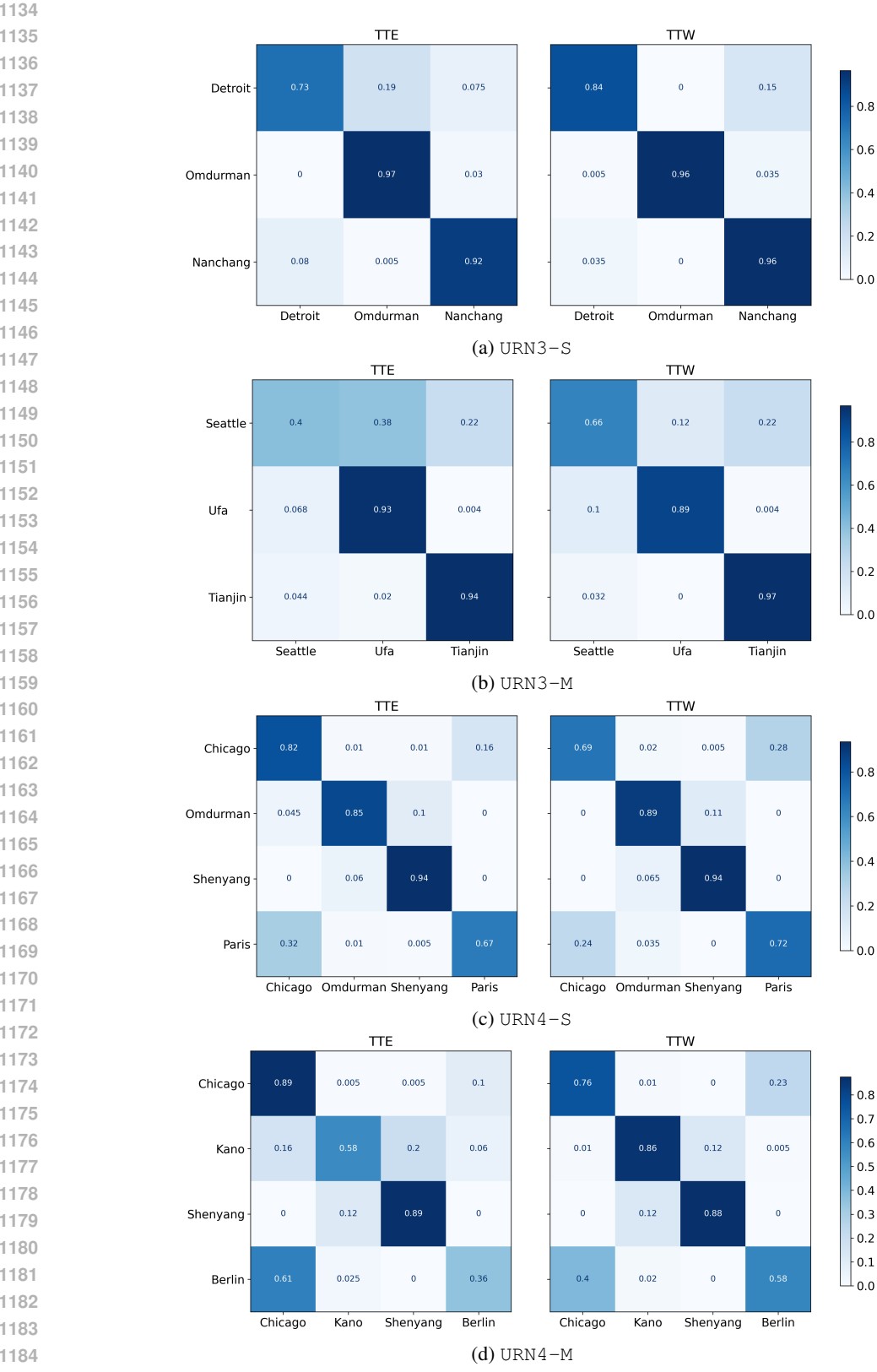

Figure 7: Confusion matrices of classification on `URN3-S` and `URN3-M`, `URN4-S` and `URN4-M`. The row indices are true labels and the column indices are predicted labels.

