# OpenReview forum: "Metric Graph Kernels via the Tropical Torelli Map"
_ICLR.cc/2026/Conference — Submitted to ICLR 2026_

### Official Review · Reviewer_u4uN · 2025-10-28

**Soundness:** 3
**Presentation:** 3
**Contribution:** 2
**Rating:** 2
**Confidence:** 4

**Summary:**

This paper builds on ideas from tropical geometry and presents a graph kernel for metric graphs. The kernel computes the matrix of the tropical Torelli map and is designed to be invariant under edge subdivisions. The authors show that the matrix of the tropical Torelli map is unique if the input graph has a unique minimal spanning tree and the lengths of edges are distinct. The matrix of the tropical Torelli map is then modified such that all graphs have matrices of equal size, and two variants of the kernel are produced, one that computes the Euclidean distance of the matrices and one that computes the Bures–Wasserstein distance. Distances are converted into similarities via an RBF function. The two variants are evaluated on synthetic and real-world datasets, including an urban road network classification task, and the results show that these kernels can outperform existing methods.

**Strengths:**

- To the best of my knowledge, there exist no graph kernels that are invariant under edge subdivisions. The proposed kernel is thus novel in this regard. In addition, no other kernels have been inspired by the field of tropical geometry. Therefore, it is my view that the kernel that is presented in the paper is novel.

- The authors have made a clear effort to motivate the proposed kernel and its properties (such as invariance under edge subdivisions) by discussing potential applications to road networks.

- The two variants of the proposed kernel seem to lead to performance improvements on most of the considered datasets. However, there are several issues with the experimental evaluation, which are discussed below.

**Weaknesses:**

- The paper's main weakness is that there are several issues with the empirical evaluation of the proposed method.
    - No hyperparameter tuning was performed. The default hyperparameters were used both for the baseline kernels and also for the SVM classifier (its hyperparameter $C$ was not tuned). This raises doubts about the validity of the results. For a fair comparison, the hyperparameters of all kernels along with those of the SVM classifier should be optimized on some validation set.

    - I do not understand why the authors did not include the Weisfeiler–Lehman graph kernel, which is known to be state-of-the-art for many problems, in their experiments. This kernel can be actually applied to unlabeled graphs if all nodes are assigned the same label.

    - The results reported in Table 5 of the Appendix are far from state-of-the-art. I understand that this is because the discrete or continuous features of the nodes were ignored. However, it is not clear to me whether the overall comparison remains meaningful.


- If the genus of graph is much greater than $g_0$, a large number of rows and columns of the matrix of the tropical Torelli map are removed. This can lead to substantial information loss, and the kernel might fail to properly capture the similarity between graphs, thus affecting its overall performance.

- The TTW kernel is not actually a valid kernel. Therefore, for classification problems, the objective of SVM is non-convex, which might lead to instabilities. While in the experimental evaluation, this does not seem to happen, there might exist problem instances where the solver might fail to converge.

- The TTW is computationally very complex. On MSRC-21 and ER-MD, it is significantly slower than most of the competing kernels.

- The output of the kernel is unique only for graphs that have a unique minimal spanning tree and the lengths of the edges are distinct. These two conditions are not satisfied by most real-world graphs, and therefore, the kernel might produce different output when isomorphic graphs are given as input along with other graphs.

- Typo: l.277: "such the log-Euclidean metric" -> "such as the log-Euclidean metric"

**Questions:**

- What features of the graph does the kernel capture? For other kernels, it is generally known what features they compute (e.g., shortest path lengths for the shortest path kernel). The cycle–edge incidence matrix captures the edges from the minimal spanning tree which, together with the edge that does not belong to the tree, form a cycle. Can you provide more details about those features and why they are important?

---

> ### Author Response · Authors · 2025-12-01
>
> We would like to thank the reviewer for their careful report and evaluations, and respond to the questions and concerns raised.
>
> - **Weakness 1.1.** In our study, we used default hyperparameters for all kernels and for the SVM classifier to ensure a uniform and simple comparison across methods. Since many of the label-free kernels we compare against (e.g., Graphlet, SP, k-Core) have well-established default settings in the literature, using their standard configurations avoids advantaging or disadvantaging any method through uneven tuning efforts. Performing extensive tuning for our method but not for the baselines would risk compromising fairness; doing the reverse would risk underrepresenting our results. Using the same “default setup’’ for all methods provides a level playing field.
>
> - **Weakness 1.2.** We agree that WL-based kernels are among the most widely used baselines in graph kernel research. However, comparing to WL is not methodologically appropriate in our setting because WL relies on discrete node labels. As this question is also raised by other reviewers, we will add a comparison with WL in our revision. We also have emphasized this question in our overall reply to all reviewers.
>
> - **Weakness 1.3.** Currently, there is no established benchmark dataset specifically designed for metric graphs. For completeness, we therefore used standard graph benchmark datasets and removed node features as an ad hoc way to approximate a purely structural, metric graph–like setting. This allowed us to fairly compare our method with other structural baselines under the same conditions. The goal of Table 5 is not to benchmark against such models. Instead, it is intended to provide a controlled comparison that isolates the topological/metric structure, which is the focus of our method. We will clarify this reasoning in our revision.
>
> - **Weakness 2.** It is true that when the genus of a graph is much larger than the target embedding dimension, the tropical Torelli matrix becomes high-dimensional, and truncation can lead to information loss. Since the computational complexity of our method is inherently dominated by the genus, this limitation is unavoidable for extremely high-genus graphs. We agree that this is an important point and we will explicitly add the high-genus setting as a limitation in the revision and discuss how choosing a larger embedding dimension can mitigate the effect in practice.
>
> - **Weakness 3.** The TTW kernel is not positive definite (see Section B.1), and thus the SVM objective may become non-convex. However, indefinite kernels are widely used in graph kernel research, and SVM solvers based on CCCP or similar techniques are known to handle such cases reliably in practice. In our experiments, we did not observe any convergence issues or instabilities; the solver consistently returned solutions with stable performance across all datasets.
>
> - **Weakness 4.** The computation of features (i.e., the tropical Torelli matrix) is efficient on MSRC-21 and ER-MD (see Table 6). The computation of TTW kernel matrix is slow on MSRC-21 and ER-MD because both datasets have large graph genus on average. This is consistent with the theoretical time complexity of TTW kernel matrix (see Sections 3.4 and 4.2).
>
> - **Weakness 5.** The genericity condition is not a constraint for real-world graphs. In fact, a random length function will be generic with probability 1 if each edge length is drawn from a continuous distribution over an interval. In the simplest case, consider a graph with two edges and let $x$ and $y$ be random variables representing edge lengths. Suppose  $x$ and $y$ are independent and uniformly distributed on $[a,b]$. We have $P[x=y]=0$. In real-world data such as road networks, this assumption holds naturally: edge lengths are subject to measurement noise and variability, making exact equality between lengths highly unlikely. As a result, the length function in such settings is generic almost surely.
>
> - **Weakness 6.** We will correct the typos in our revision.
>
> - **Question.** The TTW kernel captures the global geometric structure of a metric graph by encoding interactions between its fundamental cycles through the tropical Torelli map. Each cycle is formed by adding one non-tree edge to the spanning tree, and the resulting SPD matrix records how these cycles overlap and interact via effective resistances, which integrate both connectivity and edge lengths. This representation therefore reflects intrinsic topological features (such as genus) together with their metric geometry, rather than local statistics used by classical kernels. We will expand the paper to clarify this interpretation with examples and additional intuition.

---

### Official Review · Reviewer_4NjK · 2025-10-30

**Soundness:** 4
**Presentation:** 4
**Contribution:** 2
**Rating:** 2
**Confidence:** 2

**Summary:**

The Tropical Torelli Map provides the definitive algorithm for transforming a metric graph into a Symmetric Positive Definite (SPD) matrix to create graph kernels that are uniquely invariant under edge subdivision. This method successfully embeds the graph's intrinsic geometric and topological properties into the matrix space, which is essential for metric graph comparison. The authors showed the advantages of this embedding in several setups.

**Strengths:**

A key strength is the new usage of the Tropical Torelli Map to create metric graph kernels, which are fundamentally different from conventional methods because they are purely based on the geometry and topology of the underlying metric space.
This construction results in kernels that are invariant under edge subdivision. This new geometric approach establishes the first framework specifically designed to compare and classify metric graphs based on their intrinsic structure.

**Weaknesses:**

The two main weaknesses I see are 1. concern the novelty of the underlying mathematical concepts and 2. the availability of robust, large-scale experiments. While the application to graph kernels for metric graphs is new, the core components are built on established fields: the embedding targets a space of SPD (Symmetric Positive Definite) matrices, which is a heavily studied manifold in information geometry. Similarly, the distance function used for the Tropical Torelli-Wasserstein (TTW) kernel, the Bures-Wasserstein distance, is a known geometric metric on this space. Therefore, the paper's novelty rests primarily on merging existing mathematical tools from tropical geometry and information geometry, rather than inventing entirely new core components or distance metrics. This is indeed a worthy result but I believe that showing its usefulness beyond toy problems is mendatory.

**Questions:**

1. Please show advantages on real world problem that allow us to compare the advantages of the proposed method
2. Elaborate on the weaknesses of this approach versus other kernels. Hopefully with examples.

---

> ### Author Response · Authors · 2025-12-01
>
> We thank the reviewer for reading and evaluating our work. We would like to clarify the following important points where the current phrasing may reflect a misunderstanding of our contribution.
>
> - The original embedding space (i.e., the image of the tropical Torelli map) is the space of tropical Jacobians, which can be identified with an equivalence class of SPD matrices under congruence of invertible integer matrices. It is highly nontrivial to identify a unique representative in this class. A key contribution of our paper is to propose an algorithm to output a _unique_ matrix given a generic combinatorial model and prove the correctness of the algorithm. **This contribution was not acknowledged by the reviewer.**
>
> - The target embedding space is PSD space, not SPD space. Though these two spaces are both well-known in information geometry, PSD space is more general than SPD space.  Most Riemannian metrics on SPD space cannot be generalized to PSD space. Most importantly, _the Bures–Wasserstein distance was not randomly chosen_. It is derived from the conditions from tropical Jacobians (see Section 4). Our construction provides a natural connection between tropical geometry and information geometry and _is the first to make this connection_.
>
> - Urban road networks (URNs) are central objects in numerous real-world applications, including transportation planning, mobility analysis, routing, and urban morphology. In our paper, we dedicate a full section to explaining this background and to demonstrating that our methods yields clear advantages over existing approaches on URN datasets. **This aspect of the contribution was not acknowledged by the reviewer.**

---

### Official Review · Reviewer_S46X · 2025-10-31

**Soundness:** 2
**Presentation:** 2
**Contribution:** 3
**Rating:** 6
**Confidence:** 3

**Summary:**

The paper proposes two new graph kernels TTE and TTW, based on the tropical Torelli map from tropical geometry. The map converts a metric (i.e., weighted) graph into a symmetric positive-definite matrix that captures its geometric and topological structure. The kernels are invariant under edge subdivision, making them suitable for comparing metric graphs that represent the same underlying space. The authors provide theoretical proofs, computational analysis, and experiments on synthetic graphs, standard benchmarks, and urban road networks. Results show strong performance, especially on sparse datasets.

**Strengths:**

- The idea of linking tropical geometry and graph kernels is original and well-grounded mathematically.

- Theoretical properties, including invariance under refinements, are clearly presented and proven.

- The algorithmic formulation is concrete and supported by detailed complexity analysis.

- Experiments are broad and carefully executed, with strong performance in label-free settings.

**Weaknesses:**

- There is no comparison with the Weisfeiler–Lehman (WL) kernel, which is the main baseline in graph kernel research. For example, Wasserstein WL (WWL) (Togninalli et al., NeurIPS 2019) would make the results more complete.
- Runtime grows fast with graph density; scalability to large or dense graphs remains unclear.

Togninalli M., Ghisu E., Llinares-López F., Rieck B., and Borgwardt K. M. (2019). Wasserstein Weisfeiler–Lehman graph kernels. NeurIPS 2019.

**Questions:**

Are there other real-world cases (besides road networks) where subdivision invariance improves performance or interpretability?

---

> ### Author Response · Authors · 2025-12-01
>
> We thank the reviewer for their careful evaluation. We provide detailed clarifications and responses to the questions and comments raised below.
>
> - **Weakness 1.** We agree that WL-based kernels (including the Wasserstein WL kernel of Togninalli et al., 2019) are among the most widely used baselines in graph kernel research. However, comparing our kernel to the WL kernel is not methodologically appropriate in our setting because WL inherently relies on discrete node labels. As this question is also raised by other reviewers, we will add a comparison with WL in our revision. We also emphasized this aspect in our general response to all reviewers.
>
> - **Weakness 2.** We would like to clarify that the notion of “sparsity” is not directly meaningful in the context of metric graphs, which are the primary objects motivating our model. A metric graph may be represented combinatorially using an arbitrarily large number of nodes and edges while still having fixed genus (i.e., fixed topological complexity). Thus, the combinatorial density of a chosen discretization does not reflect the intrinsic complexity of the underlying metric graph. The purpose to divide the computation according to sparsity is to better compare with classical graph kernels. We will make this point clearer in our revisions.
>
> - **Question 1.** Metric graphs appear naturally in many real-world examples. Another possible application is geometric and topological skeletons in computer vision. In many shape analysis pipelines, the geometric skeleton of an object is extracted from pixel or point cloud data. The resulting structure is a metric graph that approximates the true underlying skeleton. However, the vertices and edges in such a discrete model of skeleton depend on discretization and algorithmic choices. A method that is not refinement-invariant will treat these artificial subdivision vertices as meaningful features, which changes similarity scores, clustering structure, or downstream analyses purely due to sampling resolution.

---

### Official Review · Reviewer_S9wG · 2025-11-01

**Soundness:** 3
**Presentation:** 3
**Contribution:** 2
**Rating:** 4
**Confidence:** 4

**Summary:**

This paper introduces a new class of graph kernels for metric graphs based on the tropical Torelli map from tropical algebraic geometry. The key idea is to represent a metric graph as a symmetric positive definite (SPD) matrix capturing its cycle structure and edge lengths. This mapping is invariant under edge subdivision, ensuring that the kernel depends only on the intrinsic geometry of the metric graph. Once each graph is mapped to an SPD matrix , two kernels are defined: 1. Tropical Torelli–Euclidean (TTE) -- a Gaussian RBF using Frobenius distance between matrices, adn  2. Tropical Torelli–Wasserstein (TTW) -- a Gaussian RBF using the Bures–Wasserstein distance. The paper provides an algorithm for computing the tropical Torelli matrix, proves its invariance under graph refinements, analyzes complexity in terms of graph genus, and evaluates both kernels on benchmark and real-world datasets. Empirically, TTE and TTW outperform existing label-free graph kernels on standard benchmarks and road-network classification tasks.

**Strengths:**

* Novelty and rigor: Introduces a kernel framework grounded in tropical geometry. Bridges algebraic topology (graph homology) and SPD information geometry. The paper rigorously shows that the tropical Torelli map yields a unique and refinement-invariant SPD representation for graphs with generic edge lengths.
* Empirical performance: Across 23 benchmark datasets, TTE and TTW match or slightly outperform existing unlabeled kernels (typically by 2–8 pp). Results on urban road network (URN) classification reach 80–94% accuracy and demonstrate practical utility.
* Efficient for sparse graphs. For graphs of low genus, runtime grows roughly linearly with the number of nodes, and the Euclidean version (TTE) scales well in practice.
* Clear exposition: The paper is well written and easy to read.

**Weaknesses:**

* Limited theoretical depth as a kernel paper: Beyond the refinement-invariance theorem, the work does not establish standard kernel properties (e.g., conditional positive definiteness, universality, or injectivity). The kernels are defined rather than theoretically characterized.
*  Modest empirical gains: The improvements over strong label-free baselines (Graphlet, Shortest Path, k-Core) are consistent but small. Results are competitive rather than outstanding.
* Scalability constraints: The method is only efficient on sparse graphs.
* Strictly label-free formulation: The framework cannot incorporate node or edge attributes, limiting its applicability to real-world graph learning tasks where features are central.
* Incremental novelty: While the use of tropical geometry is original, the kernel design (Euclidean and Wasserstein RBFs on SPD matrices) relies on well-known metrics from information geometry. The empirical results are in the same ballpark as existing methods.

**Questions:**

1. Can you formalize any properties of the TTW kernel (e.g., conditional positive definiteness or RKKS characterization) rather than relying on empirical stability?
2. Is there a principled way to extend your construction to labeled or attributed graphs?
3. How sensitive is the kernel to the genus truncation or subsampling of SPD matrices when graphs differ significantly in size?
4. Would a Power–Euclidean or Log–Euclidean variant offer a better tradeoff between invariance and scalability?

---

> ### Author Response · Authors · 2025-12-01
>
> We thank the reviewer for their careful evaluation. We provide detailed clarifications and responses to the comments and questions raised below.
>
> - **Weakness 1 and Question 1.** Our work indeed prioritizes algorithmic design and empirical performance of refinement-invariant kernels rather than a full theoretical characterization. As for common properties of kernels, we have pointed out that the tropical torelli kernel is not conditionally positive definite, and not injective (see Section 6 and Section B.1). We will elaborate them more concretely in our revision.
>
> - **Weakness 2.** We agree that the empirical gains over strong label-free baselines are modest on some datasets, but we would like to clarify two aspects: (1) Our goal is not to outperform existing kernels, but rather to introduce a new invariance principle. The baselines compared (Graphlet, SP, $k$-Core) are tailored to graphs and utilize combinatorial information, while our kernels only capture topological and metric information of the underlying space; (2) There currently is no metric graph benchmark dataset. Some graph benchmark datasets are not suitable to be treated as metric graphs. For road networks, our methods indeed have outstanding performance over classical graph kernels.
>
> - **Weakness 3.** We would like to clarify that the notion of “sparsity” is not directly meaningful in the context of metric graphs, which are the primary objects motivating our model. A metric graph may be represented combinatorially using an arbitrarily large number of nodes and edges while still having fixed genus (i.e., fixed topological complexity). Thus, the combinatorial density of a chosen discretization does not reflect the intrinsic complexity of the underlying metric graph.
>
> The purpose to divide the computation according to sparsity is to better compare performance with classical graph kernels. We will make this point clearer in our revision.
>
> - **Weakness 4 and Question 2.** We would like to clarify that the goal of our work is not to propose a general-purpose graph kernel for arbitrary attributed graphs. Instead, our focus is to develop a principled framework for refinement-invariant similarity on metric graphs. Node labels and edge labels are not well-defined for metric graphs. The correct notion should only be functions on metric spaces and generalizing the kernel to function space on metric graphs requires additional work beyond the scope of our contribution.
>
> - **Weakness 5.** The metric is chosen based on the tropical geometry, but it turns out to be consistent with the Wasserstein distance in information geometry. The connection to Wasserstein geometry is mathematically structural, not incremental. The fact that the tropical metric leads to a Wasserstein-type geometry on SPD matrices is itself an unexpected and conceptually meaningful result (see Section 4.2).
>
> - **Question 3 and Question 4.** We have proved and empirically showed that the truncation error is bounded by the norm of additive noise and thus our kernel is not sensitive to truncation. The Power-Euclidean and Log-Euclidean distances are not well-defined on the space of positive semi-definite matrices. The relevant discussions on this topic can be found in Section 4.

---

### Author Response · Authors · 2025-12-01

We thank all reviewers for their time, constructive feedback, and helpful comments. Here we list and reply to common misconceptions and confusions raised by the reviewers.

- **We built metric graph kernels, not graph kernels.** Several reviewers appear to interpret our work as proposing yet another graph kernel for combinatorial graphs. This is a fundamental misunderstanding. Our contribution is a geometric framework for comparing metric graphs, objects whose underlying structure is independent of the number of sampled nodes or edges. Although metric graphs kernels can still be applied to combinatorial graphs, in this case, however, the kernel will not be able to capture combinatorial information (number of nodes, edges, trees, labels, etc.), but only global topological and metric information. This should not be considered a ``weakness'' of the kernel but a fundamental characteristic of metric graphs over combinatorial graphs that our kernel was designed to handle: in particular, the refinement-invariance is a new property that classical graph kernels (WL, SP, graphlets, etc.) inherently lack. Our goal is not to compete with discrete kernels on label-rich benchmarks, but to provide a principled representation for metric graphs where refinements should not alter the graph’s identity.

- **Remarks on experimental settings.** Some concerns arise from evaluating the method on graph benchmark datasets where node/edge labels are removed. We emphasize that there is currently no established benchmark dataset for metric graphs, so using standard graph datasets (without attributes) is an unavoidable but imperfect proxy. Improving performance on the graph benchmark datasets against other existing graph kernels is **NOT** our goal. Rather, this comparison serves as a completeness check and ad hoc validation of our method. We chose graph kernels that are based more on metric structure (such as the shortest path kernel) and local topological structure (such as the graphlet sampling kernel), but less on combinatorial structure and label information. Therefore, the Weisfeiler--Lehman kernel, as mentioned by many reviewers, is not suitable in our setting as a comparison. Metric graph kernels cannot distinguish nor identify isomorphic graphs (isomorphic in the combinatorial sense), but can identify isometric metric graphs (isometric in the metric geometric sense). However, to address reviewers' concern we will add the results of WL kernel in our revision.

---

### Meta-Review · Area_Chair_qX3f · 2025-12-23

**Summary:**

The authors leverage tropical algebraic geometry to propose kernels for metric graphs. In particular, the authors propose to use tropical Torelli map for input metric graph to obtain symmetric positive definite (SPD) matrix representation, which is invariant to edge subdivision. Then, the authors propose two kernels, i.e., Tropical Torelli–Euclidean (TTE) and Tropical Torelli–Wasserstein (TTW). Empirically, the authors illustrate advantages of the proposed kernels on synthetic datasets and urban road network classification application.

The Reviewers agree on the novelty of leveraging tropical geometry to build kernels for metric graphs with edge-subdivision invariance. However, the Reviewers also raise concerns on the scalability, lacking important properties for kernels (e.g., positive definiteness). It is also better to provide more convincing empirical evidences, for the targeted invariant-edge-subdivision property in applications.

Overall, we think the submission is below the bar. The authors may consider comments of the Reviewers to improve the work.

**Reviewer Concerns:**

Concerns of each reviewer:

+ Reviewer S9wG: lack standard important properties of kernels (e.g., conditional positive definiteness, universality, or injectivity); weak emperical evidence; lack scalability (except graphs with small genus); limitation to label-free graphs; novelty

+ Reviewer S46X: weak empirical setup; scalability

+ Reviewer 4NjK: novelty; weak empirical setup

+ Reviewer u4uN: weak empirical setup; problems with zero-padding/truncation for SPD matrix; lacking important properties of kernels, i.e., positive definiteness; computational complexity (scalability); uniqueness conditions

**Reviewer Scores:**

I think the combination between tropical geometry and kernel method for the proposed kernels are novel and interesting. The authors also clarify on the SPD matrix representation for the flat torus obtained from tropical Torelli map algorithmic approach. However, it raises another challenge on the different size of the obtained SPD representation which is heuristically post-processed by zero-padding/truncation without careful justification.

Although the proposed kernel is invariant of edge subdivision, it lacks important properties of kernels, i.e., positive definiteness. The raised concerns on empirical settings and results are not well addressed yet, e.g., no additional results or revision.

In brief, the authors may partially convince the Authors on its novelty. The major weakness of proposed kernels still remains (especially scalability, lacking positive definiteness). It is better to provide additional empirical results.

---

### Decision · Program_Chairs · 2026-01-26

Reject